# Mitophagy Regulates the Circadian Rhythms by Degrading NR1D1 in Simulated Microgravity and Isolation Environments

**DOI:** 10.3390/ijms25094853

**Published:** 2024-04-29

**Authors:** Sihai Zhou, Xiaopeng Li, Fengji Liang, Guohua Ji, Ke Lv, Yanhong Yuan, Yujie Zhao, Na Yan, Chuanjie Zhang, Shiou Cai, Shuhui Zhang, Xu Liu, Bo Song, Lina Qu

**Affiliations:** 1Department of Pathology and Forensics, Dalian Medical University, Dalian 116044, China; zshdoct@163.com; 2State Key Laboratory of Space Medicine, China Astronaut Research and Training Center, Beijing 100094, China; lixiaopeng_acc@163.com (X.L.); freejohnnyliang@126.com (F.L.); jgh1682004@126.com (G.J.); lvke_med@aliyun.com (K.L.); yyh_yuan@163.com (Y.Y.); zhaoyujie9797@163.com (Y.Z.); yanna202204@163.com (N.Y.); zhangchuanjie96@163.com (C.Z.); caishiou19910228@163.com (S.C.); xuxu_lh@163.com (X.L.)

**Keywords:** circadian rhythms, mitophagy, NR1D1, urolithin A, suprachiasmatic nucleus, tail-suspension-and-isolation model

## Abstract

Long-term spaceflight is known to induce disruptions in circadian rhythms, which are driven by a central pacemaker located in the suprachiasmatic nucleus (SCN) of the hypothalamus, but the underlying molecular mechanisms remain unclear. Here, we developed a rat model that simulated microgravity and isolation environments through tail suspension and isolation (TSI). We found that the TSI environment imposed circadian disruptions to the core body temperature, heart rate, and locomotor-activity rhythms of rats, especially in the amplitude of these rhythms. In TSI model rats’ SCNs, the core circadian gene NR1D1 showed higher protein but not mRNA levels along with decreased BMAL1 levels, which indicated that NR1D1 could be regulated through post-translational regulation. The autophagosome marker LC3 could directly bind to NR1D1 via the LC3-interacting region (LIR) motifs and induce the degradation of NR1D1 in a mitophagy-dependent manner. Defects in mitophagy led to the reversal of NR1D1 degradation, thereby suppressing the expression of BMAL1. Mitophagy deficiency and subsequent mitochondrial dysfunction were observed in the SCN of TSI models. Urolithin A (UA), a mitophagy activator, demonstrated an ability to enhance the amplitude of core body temperature, heart rate, and locomotor-activity rhythms by prompting mitophagy induction to degrade NR1D1. Cumulatively, our results demonstrate that mitophagy exerts circadian control by regulating NR1D1 degradation, revealing mitophagy as a potential target for long-term spaceflight as well as diseases with SCN circadian disruption.

## 1. Introduction

It is well known that extended travel in deep space alters human physiology with an unknown molecular etiology [1]. The space environment leads to various pathophysiological effects in astronauts, such as bone demineralization, circadian rhythm dysregulation, sleep disturbances, cardiovascular and metabolic irregularities, and immune system dysregulation [2]. The disturbance of astronauts’ circadian rhythm is among the primary modifications observed in long-duration space missions [3]. Inadequate control of their internal circadian clock may lead to disruptions, posing risks to both work performance and overall health [4]. Therefore, it is crucial and urgent to explore the mechanisms of circadian rhythm disruption in astronauts and increase efforts to assess and mitigate potential damage to the crew.

In response to environmental variations in the diurnal cycle, mammals exhibit circadian oscillations in both behavior and physiology, including sleep–wake cycles, as well as rhythmic fluctuations in metabolic, endocrine, cardiovascular, and immune functions [5]. The disturbance of normal circadian rhythmicity is linked to various disease conditions, including neurodegenerative diseases and metabolic disorders [5,6]. In mammals, the circadian rhythms are driven by a central pacemaker located in the suprachiasmatic nucleus (SCN) of the hypothalamus with an approximate 24 h periodicity. Circadian rhythms are generated endogenously by the molecular circadian clock, primarily composed of transcription–translation feedback loops (TTFLs) [7,8]. In addition to TTFLs, circadian rhythm proteins can also be regulated through autophagic degradation [9,10]. For example, circadian protein CRY1 could be degraded timely by autophagy, a process essential for maintaining blood glucose levels [10]. *Nr1d1* is a nuclear receptor and a core clock-system gene that functions as a transcriptional repressor of *Bmal1* and BMAL1 target genes [11]. Studies have shown that NR1D1 and BMAL1 are essential in maintaining the core body temperature (CBT), heart rate (HR), and locomotor-activity rhythms [12,13]. The deletion of *Bmal1* in various tissues of mice abolished the circadian rhythm of HR and locomotor activity [13]. The genetic loss of *Nr1d1* disrupts body temperature rhythm and brown adipose tissue activity [12]. Additionally, *Nr1d1* has been demonstrated to play a crucial role in regulating the amplitude of circadian rhythms [14,15,16]. Pharmacologically targeting *Nr1d1/2* has been suggested as a potential treatment for sleep disorders and metabolic diseases [17].

To maintain normal cellular function, aged and damaged mitochondria require degradation through mitophagy, a type of selective autophagy [18]. Defects in mitophagy may lead to the accumulation of abnormal mitochondria, resulting in mitochondrial dysfunction and an increase in intracellular ROS levels [19]. The recent NASA twin study has confirmed that mitochondrial dysfunctions were observed in both mouse models and humans under spaceflight conditions, highlighting the significance of mitochondrial dysregulation as a central element in space biology, including factors such as circadian rhythms [1]. Furthermore, they discovered a widespread change in circadian rhythm pathway genes in all internal organs of mice, except for the liver, during spaceflight [1]. However, the molecular mechanisms linking circadian rhythm disruptions to mitochondrial dysregulation in the spaceflight condition are still unclear.

The on-orbit environment is complex and multifaceted, including factors such as microgravity and isolation. Tail suspension in rodents has been widely used to simulate the effects of microgravity [20,21]. Here, we established a rat model to simulate microgravity and isolation environments by tail suspension and isolation (TSI). To investigate the circadian rhythms of the rats, a VitalView^TM^ data-collection system was utilized to record the core body CBT, HR, and locomotor-activity rhythms [19]. The results indicated that the TSI environment disrupted the normal rhythms of CBT, HR, and locomotor activity in rats, particularly affecting their amplitudes. Our findings suggest that mitophagy plays a role in regulating circadian rhythms by degrading NR1D1. Furthermore, treatment with urolithin A (UA), a mitophagy activator derived from pomegranate nuts and berries [22], reversed the disruptions in CBT, HR, and locomotor activity rhythms caused by the TSI environment.

## 2. Results

### 2.1. TSI Environment Reduces the Amplitude of Circadian Rhythms of Rats

In mammals, the SCN orchestrates a multitude of circadian biological rhythms that intricately modulate a wide range of behavioral and physiological functions, including body temperature, HR, and locomotor activity [23,24]. To investigate the impact of the TSI environment on the SCN rhythms in rats, CBT, HR, and locomotor activity were recorded using transmitters over a period of 4 weeks (Figure 1A,B,H,K and Appendix A). To eliminate the influence of the TSI model on rat activity, the locomotor activity was analyzed during the initial 7 days. The results showed that the TSI model itself did not impact the total activity (Appendix A), as well as the amplitude, daytime, night-time, and mesor of activity (Appendix A). By analyzing the circadian rhythm data of rats over a period of 28 days, our results suggested that in the TSI environment, abnormal rhythms in CBT and HR might begin to appear after 10–12 days and stabilize in 21–28 days (Appendix A). Therefore, further analysis was performed for the final 7 days. The data showed a significant decrease in the amplitude of CBT, HR, and locomotor-activity rhythms in the TSI group (Figure 1C,I,L). Additionally, a reduction in CBT levels during daytime, night-time, and mesor was observed (Figure 1D–F). Moreover, a circadian phase advance in CBT was noted in the model group (Figure 1G), along with an increase in HR and locomotor-activity levels during the daytime (Figure 1J,M), as well as a significant rise in locomotor activity during the night-time and mesor (Figure 1N,O). The attenuated 24 h period (tau) in the TSI group’s HR due to the dramatic decrease in amplitude was also observed (Figure 1H), while there were no changes in the period of CBT and locomotor activity (Appendix A). Taken together, these results suggest that simulating microgravity and isolation environments through TSI could contribute to the disruption of CBT, HR, and locomotor-activity rhythms in rats, particularly affecting circadian amplitude.

### 2.2. TSI Environment Leads to the Aberrant Expression of NR1D1 and BMAL1 in SCN Tissues

The proper functioning of circadian rhythms depends not only on the period and phase, but also on the adjustment of amplitude. Low-amplitude molecular oscillations are insufficient to sustain robust circadian physiology. Our results showed a remarkable decrease in the amplitude of CBT, HR, and locomotor activity rhythms, particularly in the amplitude of HR rhythms. Previous study has indicated that *Nr1d1* plays a crucial role in regulating the amplitude of clock transcription [14,15,16]. Moreover, *Nr1d1* and its target gene Bmal1 are essential to maintain body temperature, HR, and activity rhythms [12,13,25]. To gain further insights into the cause of rhythm disturbance, we collected SCN tissues from rats in two groups at 4 h intervals over a 24 h period (Figure 2A). Compared to the control group, a significant upregulation of the NR1D1 protein was detected in TSI rats at ZT3 and ZT11, ZT15, and ZT23 (Figure 2B,C), while we detected no changes in its mRNA levels (Figure 2F). In addition, TSI suppressed the protein expression of the BMAL1 at ZT3, ZT7, and ZT11 (Figure 2B,D). Moreover, we also observed a significant decrease in the amplitude of NR1D1 and BMAL1 protein levels (amplitude is defined as half the oscillation); (peak value − trough value)/2) (Figure 2E). These findings suggest that the aberrant expression of NR1D1 and BMAL1 in SCN tissues may contribute to the disruption of circadian rhythms observed in TSI groups.

### 2.3. Autophagy Degrades NR1D1 through Directly Binding to Its LIR Motifs

In addition to transcriptional regulation, the maintenance of proper circadian rhythms depends on the timely degradation of clock-related proteins [26]. Certain clock proteins are degraded through autophagy pathways [10,27]. Our study revealed that the upregulation of the NR1D1 protein was accompanied by no change in mRNA levels, leading us to postulate that the observed increase in NR1D1 might be attributed to impaired autophagy-dependent degradation. To confirm this hypothesis, we first used siRNA targeting *Atg5* to block autophagy in NIH3T3 cells. The results showed that the knockdown of ATG5 resulted in enhanced NR1D1 expression and decreased BMAL1 protein levels (Figure 3A,B). The treatment of chloroquine (CQ), a lysosomal inhibitor, also induced a similar change of NR1D1 and BMAL1 expression (Appendix A). Autophagic and proteasomal degradation systems are two major quality-control pathways responsible for maintaining cellular homeostasis [28]. In this study, we inhibited the proteasome pathway using MG132 and examined its effects on NR1D1. The results showed that MG132 had no effect on the protein expression of NR1D1 and BMAL1 (Appendix A). Those findings suggest that NR1D1 may be degraded through autophagy.

Typically, autophagic substrates contain one or more LC3-interacting region (LIR) motifs, which facilitate their interaction with LC3 and their subsequent degradation via the autophagy–lysosome pathway [29]. To further investigate how autophagy regulates NR1D1, we analyzed the potential motifs targeted by LC3. The canonical LIR motifs consist of W/F/Y-X-X-I/L/V sequences [29]. Through the iLIR web server [30], we identified two highly predicted candidate LIR motifs within NR1D1, which exhibit conservation among mammalian species (Figure 3C). An immunofluorescence assay showed the colocalization of endogenous LC3 with NR1D1 in NIH3T3 cells and the SCN tissues of rats (Figure 3D). The immunoprecipitation assay confirmed the direct interaction between NR1D1 and LC3 (Figure 3E). Through mutated NR1D1 plasmids (Figure 3F), we then investigated whether two LIR motif mutations attenuated the degradation of NR1D1. As indicated, NIH3T3 cells transfected with mLIR1 and mLIR2 plasmids showed a notable restoration in NR1D1 protein levels, respectively, indicating the necessity of the two LIRs in the degradation of NR1D1 by LC3 (Figure 3G,H), and this restoration was accompanied by a significant reduction in the expression of the BMAL1 protein (Figure 3G,H). In addition, we observed a similar change in the expression of NR1D1 and BMAL1 in NIH3T3 cells upon the activation of autophagy through UA, when transfected with either mLIR1 or mLIR2 plasmids, or a combination of both (Figure 3I,J). Taken together, our data suggest that NR1D1 can be degraded by autophagy through directly binding to the autophagosome marker LC3 via two NR1D1 LIR motifs.

### 2.4. TSI Environment Causes Mitochondrial Dysfunction and Mitophagy Deficiency in Neurons of SCN

Mitochondrial dysfunction is one of the fundamental biological responses to spaceflight travel, accompanied by a decrease in mitochondrial membrane potential and unbalanced redox homeostasis [1,31]. Our previous results indicated significant changes in the expression of NR1D1 and BMAL1 proteins in rat SCN tissue at both ZT3 and ZT11 under TSI conditions (Figure 2B–D). Therefore, we chose these two time points for subsequent experiments. We first investigated whether mitochondrial morphology disorders occurred in SCN neurons in TSI models using transmission electron microscopy (TEM). As predicted, an abnormal mitochondrial ultrastructure was shown within the model group. Notably, certain mitochondria exhibited significant swelling and vacuolation, along with an evident disorganization and disruption of cristae (Figure 4A,B). Moreover, we also observed a significant reduction in mitochondrial membrane potential (Figure 4C), along with a decreased activity of superoxide dismutase (SOD) and an increased level of malondialdehyde (MDA) in the SCN (Appendix A). Mitophagy, a specialized form of autophagy, plays a crucial role in maintaining cellular homeostasis and mitochondrial quality by selectively eliminating damaged or dysfunctional mitochondria [18]. To investigate whether mitophagy was involved, the number of mitophagies was measured. TEM data demonstrated that TSI led to a significant loss of mitophagic-like structures (Figure 4D,E). Accordingly, Western blotting confirmed the decreased mitophagy in the model group, as evidenced by corresponding changes in protein levels of p62, LC3II/I, and TOMM20 at ZT3 and ZT11 (Appendix A). Mitochondrial mass is controlled by the equilibrium between the mitochondria biogenesis and the degradation of existing mitochondria via mitophagy [32]. Thus, we also observed an increase in mitochondrial content as measured with the mtDNA/nDNA ratio (Figure 4F). These results demonstrate that the TSI environment induces mitochondrial dysfunction and mitophagy deficiency in the SCNs of rats. Our findings revealed that NR1D1 could be degraded by autophagy (Figure 3). Given that mitophagy constitutes a specialized branch of autophagy, we were particularly interested in investigating whether it could degrade NR1D1. UA, an activator of mitophagy, significantly promoted the degradation of the NR1D1 protein while simultaneously restoring the level of BMAL1 proteins (Figure 4G,H). Conversely, Mdivi-1, a mitophagy inhibitor, suppressed the degradation of the NR1D1 protein and led to decreased levels of BMAL1 proteins (Figure 4I,J). These findings suggest that NR1D1 may also undergo degradation via mitophagy. In conclusion, the data above indicate that mitophagy deficiency induces mitochondrial dysfunction, and an abnormal expression of NR1D1/BMAL1 may account for circadian rhythm disturbances in rats subjected to the TSI model.

### 2.5. UA Ameliorates the Disturbance of SCN Rhythms

UA enhances cellular mitochondrial health by activating mitophagy, which has been demonstrated to have beneficial effects on various organs such as muscles, the brain, and joints [22]. Therefore, we sought to investigate whether a 28-day period of UA treatment (20 mg/kg/d) could counteract the circadian rhythm disturbances caused by the TSI environment. Our results revealed that the UA-treated groups exhibited 24 h circadian oscillations in CBT, HR, and locomotor-activity rhythms, effectively mitigating the disruption of these rhythms throughout the entire 28-day period (Figure 5A,F,K, and Appendix A). In comparison to the TSI group, UA treatment significantly increased the amplitude of CBT, HR, and locomotor-activity rhythms (Figure 5B,G,L). Furthermore, we observed elevated CBT levels during both daytime and night-time, as well as an increase in the mesor (Figure 5C–E). The UA-treated groups showed significantly reduced HR values during daytime periods (Figure 5H). Additionally, a significant decrease in activity levels during the daytime and mesor was observed in the UA-treated groups (Figure 5M,N). In summary, our findings suggest that UA may alleviate SCN rhythm disruptions induced by the TSI environment, highlighting its potential as a therapeutic intervention for circadian rhythm disturbances.

### 2.6. UA Mitigates SCN Rhythm Disruption and Mitochondrial Dysfunction through Activating Mitophagy

To further investigate the impact of UA on SCN neurons, rat SCN tissues were collected following a 4-week treatment with UA under TSI conditions at ZT3 and ZT11. Our results indicated that UA treatment significantly enhanced mitophagy in SCN tissues compared to the TSI groups (Figure 6A,B and Appendix A). Subsequent analysis revealed a marked reduction in the quantity of abnormal mitochondria (Figure 6C,D), accompanied by increased SOD activity and decreased MDA levels in SCN tissues from UA-treated groups (Appendix A). Furthermore, we observed a downregulation in NR1D1 expression and an upregulation in BMAL1 expression in the SCN tissues of UA-treated groups (Figure 6E–H). Collectively, our findings suggest that UA may regulate the expression of NR1D1 and BMAL1 in SCN tissues, and it alleviates SCN rhythm disruption and mitochondrial dysfunction through the activation of mitophagy.

## 3. Discussion

Spaceflight is a complex environment that can exacerbate chronic inflammation, disrupt circadian rhythms, and induce stress responses [33]. Hindlimb unloading (HU) through tail suspension in rodents has been widely used to simulate the effects of microgravity [20,21]. Notably, HU is acknowledged for triggering a stress response, which has the potential to disturb circadian rhythms [34,35], suggesting that stress responses induced by HU in rats could influence circadian rhythms. Nevertheless, the study revealed that the level of blood corticosterone (CORT), a stress indicator, did not exhibit significant alterations following a 7-day period of HU [36]. Our observations on CBT, HR, and locomotor-activity rhythms indicated subtle changes in circadian rhythms during the initial 7 days. Abnormalities in CBT and HR rhythms may begin to manifest after 10–12 days, stabilizing between days 21 and 28 in the TSI environment (Appendix A and Figure 1K). These findings suggest that while TSI environments may trigger stress responses in rats, prolonged exposure leads to adaptation, resulting in a relatively minor impact of stress responses on circadian rhythms over the 28-day period.

To elucidate the molecular mechanisms behind circadian rhythm disruptions in the on-orbit environment, we developed a rat model to simulate microgravity: isolation conditions using TSI. Mitophagy, a type of selective autophagy, functions to remove damaged mitochondria. The relation between the circadian clock and mitophagy has been reported in several studies. A loss of *Clock* gene activity both in vitro and in vivo leads to increased cardiac injury and ventricular dysfunction due to the impaired activation of autophagy/mitophagy [37]. NR1D1 regulates BNIP3-mediated mitophagy in ulcerative colitis to alleviate colitis symptoms [38]. The knockdown of HDAC3 in the heart could alleviate diabetic myocardial infarction/reperfusion injury by modulating the NR1D1/BMAL1 pathway to activate mitophagy [39]. Our study reveals a novel role of mitophagy in degrading the core circadian protein NR1D1 through two LIR motifs, thereby regulating circadian rhythms and mitigating oxidative stress and mitochondrial dysfunction in the SCN of rats within the TSI environment.

The core circadian genes *Nr1d1* and *Bmal1* have been found to be essential in regulating various physiological phenotypes, including HR, blood pressure, body temperature, and locomotor activity [12,13,40]. Studies have shown that the disruption of the amplitude of body temperature rhythms occurs when *Nr1d1* is genetically lost in brown adipose tissue (BAT) [12]. Interestingly, the deletion of both *Nr1d1* and *Nr1d2* in BAT allows mice to effectively regulate their body temperature even under chronic cold conditions [25]. These findings highlight the crucial role of *Nr1d1* in the oscillation of CBT rhythms. Furthermore, NR1D1 has been shown to play a critical role in regulating the amplitude of circadian rhythms [14,15,16]. Pharmacologically targeting the circadian rhythm through synthetic NR1D1 ligands may prove to be beneficial in the treatment of sleep disorders and metabolic diseases [17]. Additionally, the deletion of Bmal1 in various tissues of mice eliminates the circadian rhythm of HR and locomotor activity [13]. Microgravity has a broad impact on the circadian clock. A multi-omics analysis reveals that circadian rhythm pathways were upregulated at the transcriptional level in all internal organs of mice during spaceflight, except for the liver [1]. Research has demonstrated that simulated microgravity through tail suspension significantly attenuated the diurnal variations of the circadian gene *Bmal1* in the rats’ SCNs and cerebral arteries [41]. In this study, we found that the TSI environment abolished the normal rhythms of rats’ CBT, HR, and locomotor activity, with a particular impact on their amplitudes. Our results revealed an elevated expression of NR1D1, predominantly at the protein level rather than the mRNA level. Concurrently, there was a decrease in the expression of BMAL1 in the SCN of the hypothalamus, particularly during ZT3 and ZT11. These findings underscore the crucial roles of NR1D1 and BMAL1 in response to the TSI environment.

NR1D1 is primarily regulated by its own transcriptional feedback loop within the circadian clock machinery, known as the TTFLs. Additionally, NR1D1 expression can be influenced by various external factors, such as light–dark cycles, hormones, and metabolic signals, which further modulate its activity and impact downstream gene expression [16]. Moreover, apart from the TTFLs, NR1D1 can also be regulated through post-translational modifications and degradation. Previous studies have demonstrated that NR1D1 is phosphorylated by CDK1, leading to its subsequent degradation via the proteasome targeted by FBXW7. The deletion of FBXW7 significantly diminishes the circadian clock amplitude [14]. Another study highlighted that the E3 ligase Siah2 regulates the degradation of NR1D1, and the depletion of Siah2 affects the period length of the circadian clock [42]. These findings underscore the importance of regulating NR1D1 degradation for maintaining proper time-keeping functionality. In our study, we discovered that NR1D1 could be degraded by mitophagy through its two LIR motifs. Furthermore, our results demonstrated that the degradation of NR1D1 occurs through ATG5- and lysosome-dependent mitophagy. Consistent with our in vitro findings, we observed significant mitophagy deficiency in the SCN tissues of rats in the TSI group. This deficiency of mitophagy in the SCN may lead to an elevated expression of NR1D1 in the TSI group, ultimately resulting in disturbances to circadian rhythms. Although we found that mitophagy defects contribute to the disruptions in circadian rhythms, our study lacks experiments that elucidate the relative contribution of the mitophagy to the degradation of NR1D1.

UA, a metabolite synthesized by the gut microbiota during the digestive process of ellagitannins, represents a class of polyphenols present in select fruits and nuts such as pomegranates, strawberries, raspberries, walnuts, and oak-aged red wine [22,43]. One of the most consistent effects of UA observed across multiple species is the improvement of mitochondrial health, as demonstrated by its role as a mitophagy agonist in diverse organisms [22]. Furthermore, preclinical studies have revealed its anti-inflammatory, antioxidant, anti-carcinogenic, and lifespan-prolonging properties [44,45,46,47]. Ongoing research is exploring the potential applications of UA in ameliorating age-related muscle decline and preventing a range of disorders such as neurodegenerative conditions, metabolic syndrome, and cancer [22]. We performed UA administration under TSI conditions to examine its role in regulating SCN rhythms and functions. Our findings suggest that activating mitophagy through UA treatment has the potential to alleviate disruptions in circadian rhythms related to CBT, HR, and locomotor-activity rhythms, as well as improving mitochondrial function and mitophagy in SCN tissues. These data unveil a novel role of UA in modulating circadian rhythms and rectifying mitochondrial dysfunction by promoting mitophagy.

In conclusion, our study reveals that mitophagy defects in rat SCN tissues, caused by the TSI environment, lead to disruptions in CBT, HR, and locomotor-activity rhythms. The core circadian protein NR1D1 plays an essential role in maintaining the amplitude of these rhythms. NR1D1 could be degraded via mitophagy by directly binding to the autophagosome marker LC3 through two NR1D1 LIR motifs. The activation of mitophagy using UA was found to effectively alleviate the disruption of circadian rhythms in rats within the TSI environment. Cumulatively, our results demonstrate that mitophagy exerts circadian control by regulating NR1D1 degradation, revealing mitophagy as a potential target for long-term spaceflight as well as diseases with SCN circadian disruption.

## 4. Materials and Methods

### 4.1. Animals and Cell Lines

All experimental procedures conducted on Sprague Dawley (SD) rats for this study were approved by the Animal Care and Use Committee of the China Astronaut Research and Training Center. The rats were housed under a 12 h light–dark cycle, with lights on at 7 a.m. (referred to as Zeitgeber time point 0, ZT0) and off at 7 p.m. (referred to as Zeitgeber time point 12, ZT12). All procedures were performed in accordance with the guidelines for the use and care of live animals set forth by the Committees of Animal Ethics and Experimental Safety of the China Astronaut Research and Training Center (reference number: ACC-IACUC-2022-016) and were duly authorized. NIH3T3 cells (ATCC, USA) were cultured in DMEM (Gibco, Waltham, MA, USA) containing 10% fetal bovine serum (FBS; Gibco, Waltham, MA, USA), penicillin (100 U/mL), streptomycin (100 μg/mL; Gibco, Waltham, MA, USA).

### 4.2. VitalView^TM^ Data-Acquisition System and Tail-Suspension-and-Isolation (TSI) Model

To investigate the impact of TSI on the physiological and phenotypic rhythms of SD rats, we utilized the VitalViewTM (Mini Mitter, Bend, OR, USA) data-acquisition system to continuously monitor the CBT, HR, and locomotor activity of the rats in real-time. Prior to tail suspension, the rats underwent implantation procedures following the manufacturer’s protocol. To simulate the effects of microgravity, 6-month-old rats were subjected to a head-down tilt of 30 degrees by suspending their tails. Each rat was housed individually in an isolated environment enclosed by ground glass. The TSI model does not affect the activity of the forelimbs in rats; they can still easily access food and water throughout the duration of the experiment, as per standard laboratory protocol, ad libitum.

### 4.3. Reagents and UA Administration

UA, CQ, MG-132, and Mdivi-1 (HY-100599, HY-17589A, HY-13259, and HY-15886, respectively) were procured from MedChemExpress (USA) and dissolved in dimethyl sulfoxide (DMSO; Sigma-Aldrich, St. Louis, MO, USA). NIH3T3 cells were exposed to UA (25 μM, 50 μM, 100 μM, or a vehicle) or mdivi-1 (12.5 μM, 25 μM, 50 μM, or a vehicle) for 48 h, or to CQ (25 μM, 50 μM, 100 μM, or a vehicle) or MG132 (5 μM) for 24 h. Subsequently, the cells were harvested for Western blotting analysis.

For the administration of UA via gavage, the rats received a daily gavage of either 20 mg/kg/d UA in PBS or PBS alone at 10 am for a duration of 28 days. After the 28-day period, the rats were euthanized, and their SCN tissues were collected for further analysis.

### 4.4. Immunofluorescence Microscopy

NIH3T3 cells were subjected to fixation with 4% formaldehyde at 37 °C for 20 min. Subsequently, the cells were washed three times for 5 min each with 0.1M PBS (Gibco, Waltham, MA, USA). Following that, NIH3T3 cells were treated with 0.5% Triton X-100 (Solarbio, Beijing, China) for 20 min, blocked with BSA for 30 min, and incubated overnight at 4 °C with primary antibodies NR1D1 (1:100; Proteintech, 14506-1-AP, Wuhan, China) and LC3 (1:100; Cell Signaling Technology, 83506, Danvers, MA, USA). The next day, the cells were washed three times with PBS. Subsequently, NIH3T3 cells were incubated at room temperature for 30 min with fluorescence-conjugated secondary antibodies (1:3000; Proteintech, SA00013-3 and SA00013-2, Wuhan, China). Following another three washes with PBS, the cells were incubated with DAPI (1:4000; cell signaling Technology; 4083, Danvers, MA, USA) in PBS for 5 min, followed by a final wash with PBS. Cell images were acquired using a LSM 710 Zeiss confocal microscope.

### 4.5. SCN Sections Immunofluorescence Microscopy

The brains were fixed in 4% paraformaldehyde in 0.1M phosphate buffer for 4 h at room temperature, with gentle shaking. Fixed brains were then cryoprotected in 10%, 20%, and 30% sucrose in PBS at 4 °C overnight. The brains were then frozen in dry ice, and coronal sections (40 μm thick) were collected using a freezing sliding microtome. The sections containing the SCN were stained following the immunofluorescence protocol.

### 4.6. Real-Time PCR

Total RNA was extracted from hypothalamic SCN tissues using TRIzol reagent (Invitrogen, Waltham, MA, USA) according to the manufacturer’s protocol. Subsequently, the RNA was reverse-transcribed into cDNA, and real-time quantitative PCR (RT-qPCR) was performed using a SYBR Green PCR Kit (Takara, Kyoto, Japan) on a light cycler (Eppendorf, Hamburg, Germany). The primer sequences of the circadian genes used in this study are provided in Table 1. The transcription levels of these genes were analyzed by employing the Comparative CT (ΔΔCT) method.

### 4.7. Western Blotting

SCN tissues from rats and NIH3T3 cells were lysed in RIPA buffer (Huaxingbio, Beijing, China) containing protease inhibitor (Aoqing Biotech, Beijing, China) and phosphatase inhibitor cocktails (Aoqing Biotech, Beijing, China) on ice for 30 min. Each protein was isolated by centrifugation at 12,000× *g* at 4°C for 20 min. The 30 μg protein was then subjected to 12.5% or 7.5% SDS-PAGE and transferred to a 0.2 μm PVDF membrane. The membranes were blocked with 5% skimmed milk and incubated overnight at 4 °C with specific primary antibodies. The primary antibodies used were as follows: NR1D1 (1:1000; Cell Signaling Technology, 13418, Danvers, MA, USA), GAPDH (1:1000; Proteintech, 60004-1-Ig, Wuhan, China), BMAL1 (1:1000; Cell Signaling Technology, 14020, Danvers, MA, USA), LC3 (1:1000; Cell Signaling Technology, 83506, Danvers, MA, USA), TOM20 (1:1000; Proteintech; 11802-1-AP, Wuhan, China), ATG5 (1:1000; Cell Signaling Technology, 12994, Danvers, MA, USA), and p62 (1:1000; Cell Signaling Technology, 23214, Danvers, MA, USA). For secondary antibodies, HRP-conjugated affinipure goat anti-mouse or anti-rabbit IgG(H + L) (1:10,000; Proteintech; SA00001-1 and SA00001-2, Wuhan, China) were used.

### 4.8. Immunoprecipitation

NIH3T3 cells were lysed in RIPA buffer (Huaxingbio, Beijing, China) containing protease inhibitors (Aoqing Biotech, Beijing, China) and phosphatase inhibitor cocktails (Aoqing Biotech, Beijing, China) for 30 min on ice. After centrifugation at 12,000× *g* at 4 °C for 20 min, the total proteins were immunoprecipitated overnight at 4 °C with LC3 antibody (1:100; Cell Signaling Technology, 83506, Danvers, MA, USA) or IgG antibody (1:100; Cell Signaling Technology, 2729, Danvers, MA, USA). The next day, 50 μL of protein A/G agarose beads (Thermo Fisher, 88802, Waltham, MA, USA) was added to the samples and incubated at 4 °C for 4 h. The precipitants were then washed 5 times with RIPA buffer, reserving 40 μL for the final wash. Subsequently, the samples were boiled with a loading buffer for 5 min and analyzed by Western blotting.

### 4.9. Transmission Electron Microscopy

The SCN tissues were cut into approximately 1 mm^3^ fragments and fixed in 2.5% glutaraldehyde (Solarbio P1126, Beijing, China) at 4 °C overnight. The samples were then washed three times for 15 min each with 0.1M PBS (Gibco, Waltham, MA, USA). After fixation with 1% osmic acid for 2 h, the tissues were washed again three times for 15 min each with 0.1M PBS. Next, the tissues were dehydrated in a gradient series of ethanol and embedded in acrylic resin at 70 °C overnight. Finally, the samples were polymerized using epoxy resin, treated at 80 °C for 24 h, then cut into ultra-thin sections of 100 nm. These sections were stained with uranyl acetate and lead citrate and observed under transmission electron microscopy (Jeol, Kyoto, Japan).

### 4.10. SiRNA Transient Transfection

The NIH3T3 cells were transfected with gene-specific siRNAs (GenePharma, Suzhou, China) and negative control siRNAs (GenePharma, Suzhou, China) using Lipofectamine RNAiMAX (Invitrogen, 13778500, Waltham, MA, USA) according to the manufacturer’s instructions. The following siRNA sequences were used: Atg5-1 siRNA (mouse: 5′-GGCAUUAUCCAAUUGGUUUTT-3′), Atg5-2 siRNA (mouse: 5′-GACGUUGGUAACUGACAAATT-3′), and nontargeting control siRNA (mouse: 5′-UUCUCCGAACGUGUCACGUTT-3′).

### 4.11. Plasmids

The NR1D1 mLIR1 and mLIR2 plasmids were created through site-directed mutagenesis (Hunan Fenghui Biotechnology, Changsha, China), and the specific LIR mutations were validated through sequencing. The NIH3T3 cells were transfected with gene-specific plasmids using Lipofectamine 3000 (Invitrogen) according to the manufacturer’s instructions.

### 4.12. The T-SOD and MDA Activity of SCN

The T-SOD activity of SCN was assessed using a Total Superoxide Dismutase Assay Kit with WST-8 (Beyotime, Beijing, China). The MDA activity of SCN was measured using a Malondialdehyde (MDA) assay kit (Beyotime, Beijing, China). All procedures were conducted according to the manufacturer’s protocol.

### 4.13. Statistical and Rhythms Analysis

All graphics and statistical calculations were generated using GraphPad Prism v8 software. For comparisons of the means of two groups, a Student’s two-tailed t-test was used and Sidak’s test was used for experiments with multiple comparisons. The significance of rhythmicity (daytime and night-time values, medians, amplitudes, and phases) was accomplished using Chronos-Fit (Chronos-Fit 1.06: Program for detection of rhythmic organization in arbitrary data. 2020. https://chronos-fit.software.informer.com accessed on 3 December 2021) [48]. * *p* < 0.05, ** *p* < 0.01, and *** *p* < 0.001 versus the corresponding controls were indicated. All values were obtained from at least three independent experiments.

## Figures and Tables

**Figure 1 ijms-25-04853-f001:**
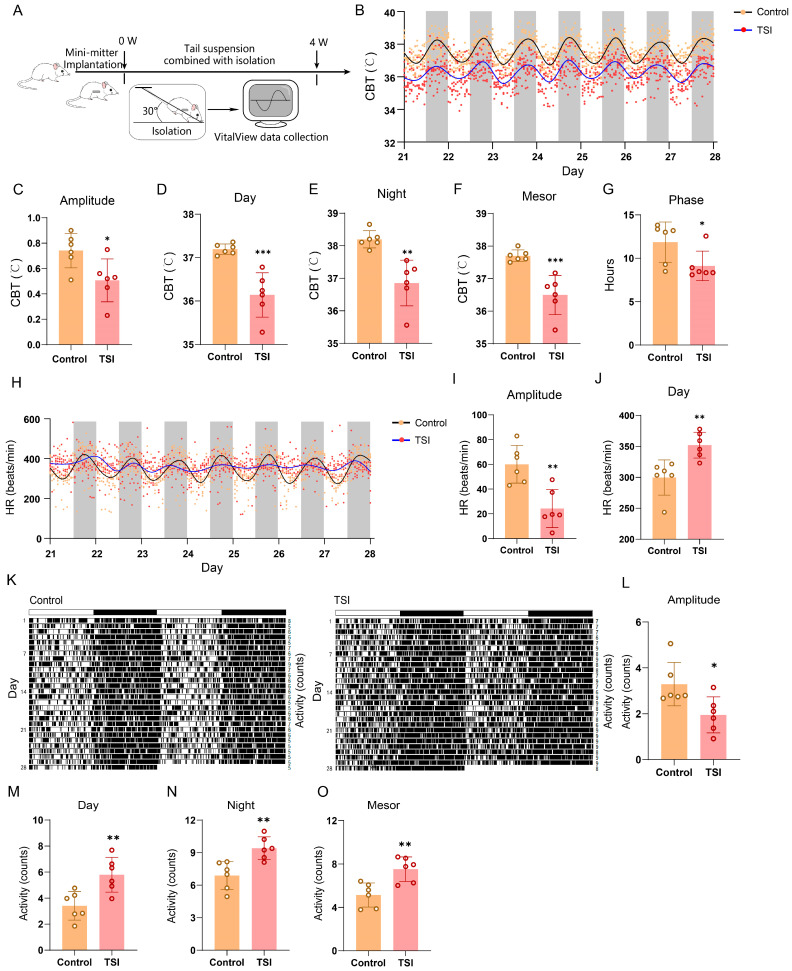
TSI induces the disturbance of circadian rhythms in rats. (**A**) Illustration of the experimental setup. Each rat was individually housed in an isolated environment and exposed to a 30-degree head-down tilt by tail suspension for 4 weeks. (**B**) The curve fitting of the CBT data of rats during the final 7-day period from control and TSI groups (each point represents the average CBT values corresponding to each hourly time point from control and TSI rats). (**C**–**G**) Analysis of amplitude (**C**), daytime (**D**), night-time (**E**), mesor (**F**), and trough phase (**G**) of the CBT data during the final 7-day period. (**H**) The curve fitting of the HR data of rats during the final 7-day period from control and TSI groups (each point represents the average HR values corresponding to each hourly time point from control and TSI rats). (**I**,**J**) Analysis of amplitude (**I**) and daytime (**J**) of the HR data during the final 7-day period. (**K**) Representative double-plotted actograms of locomotor activity of rats during the 28-day period from control and TSI groups. (**L**–**O**) Analysis of amplitude (**L**), daytime (**M**), night-time (**N**), and mesor (**O**) of the locomotor-activity data during the final 7-day period. * *p* < 0.05; ** *p* < 0.01; *** *p* < 0.001. Data are mean ± S.E.M., n = 6. Two-sided Student’s *t* tests were used to measure the significance.

**Figure 2 ijms-25-04853-f002:**
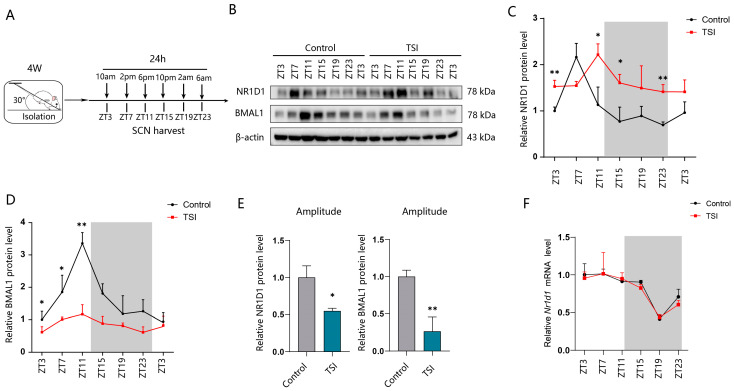
NR1D1 and BMAL1 components possibly contribute to disrupting the circadian rhythms in TSI environment of rats. (**A**) Schematic representation indicating the time point of tissue collection from rat SCN. (**B**) Western blotting analysis of the NR1D1 and BMAL1 protein in SCN samples collected at different time points from control and model groups. (**C**,**D**) Quantification for NR1D1 and BMAL1 protein expression (n = 4 rat/time/group). (**E**) Analyzing the amplitude of normalized NR1D1 and BMAL1 protein levels. (**F**) Quantifying mRNA expression levels of *Nr1d1* in SCN tissues obtained from control and model groups (n = 4 rat/time/group). * *p* < 0.05; ** *p* < 0.01. Data are means ± S.E.M. Two-sided Student’s *t* tests were used to measure the significance.

**Figure 3 ijms-25-04853-f003:**
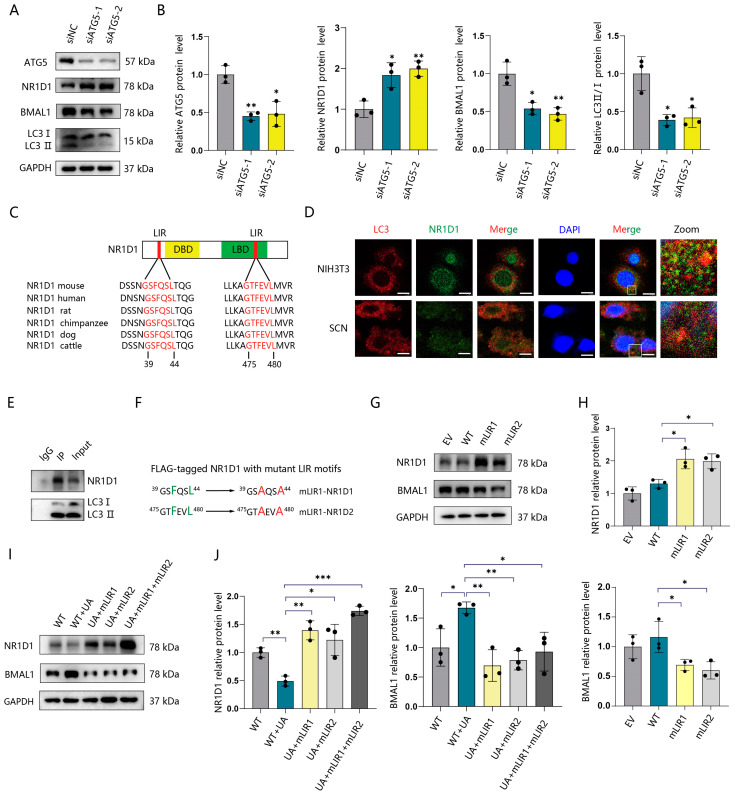
Autophagy degrades NR1D1 through directly binding to its LIR motifs. (**A**) Western blotting analysis of the proteins ATG5, LC3, NR1D1, and BMAL1 after silencing *Atg5* using interfering RNA (siRNA) in NIH3T3 cells for 48 h. (**B**) Quantification for ATG5, LC3 II/I, NR1D1, and BMAL1 protein expression (n = 3). (**C**) Diagram illustrating the domains of NR1D1 and its LIR motifs (in red) with alignments displayed across various species. DBD, DNA-binding domain; LBD, ligand binding domain. (**D**) Immunofluorescence assay analysis of endogenous LC3 colocalization with NR1D1 in NIH3T3 cells and the SCN tissues of rats. Scale bars, 15 μm. (**E**) Immunoprecipitation assay of LC3 with NR1D1 from NIH3T3 cell lysates. (**F**) Inactivation of two LIR motifs on NR1D1 through mutagenesis of phenylalanine (F) and leucine (L) residues (shown in green) to alanine (A) (shown in red). (**G**) Western blotting analysis of the proteins NR1D1 and BMAL1 in NIH3T3 cells transfected with empty vectors (EVs), wild types (WTs), and transfected with mLIR1 or mLIR2 plasmids for 48 h. (**H**) Quantification for NR1D1 and BMAL1 protein expressions (n = 3). (**I**) Western blotting analysis of the proteins NR1D1 and BMAL1 in NIH3T3 cells transfected with mLIR1 and mLIR2 plasmids under treatment with 50 μM UA or a vehicle (0.1% DMSO). (**J**) Quantification for NR1D1 and BMAL1 protein expressions (n = 3). * *p* < 0.05; ** *p* < 0.01; *** *p* < 0.001. Data are means ± S.E.M. Two-sided Student’s *t* tests were used to measure the significance.

**Figure 4 ijms-25-04853-f004:**
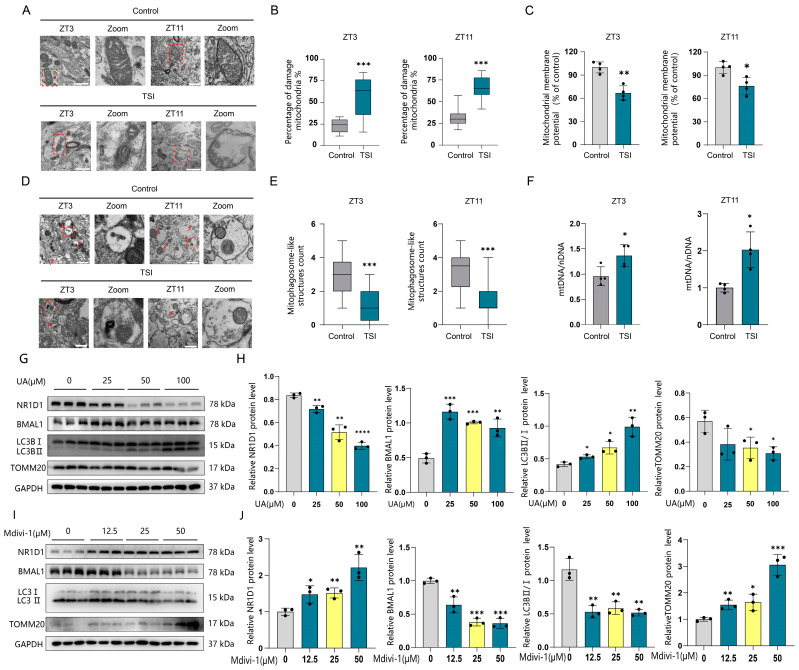
TSI environment causes mitophagy deficiency and mitochondrial dysfunction in neurons of SCN. (**A**) TEM images show the mitochondria in neurons of the SCN from both the control and model groups at ZT3 and ZT11 (n = 4). The red dashed square indicates the zoom region. Scale bars: 1 μm. (**B**) Quantification of damaged mitochondria relative to total mitochondria. (**C**) Mitochondrial membrane potential analysis conducted in the SCN tissues of the control and model groups at ZT3 and ZT11 (n = 4). (**D**) TEM images show mitophagy in neurons of the SCN from the control and model groups at ZT3 and ZT11 (n = 4). Mitophagosome-like structures are indicated by red arrowheads. The red dashed square indicates the zoom region. Scale bars: 1 μm. (**E**) Quantification of mitophagy numbers. (**F**) qPCR analysis depicts the mtDNA/nDNA ratio in the SCN of the control and model groups at ZT3 and ZT11 (n = 4). (**G**) Western blotting analysis showcases the protein expression of NR1D1, BMAL1, TOMM20, and LC3 II/I after cells were treated with UA (25 μM, 50 μM, 100 μM, or a vehicle) for 48 h (n = 3). (**H**) Quantification of NR1D1, BMAL1, TOMM20, and LC3 II/I protein expression. (**I**) Western blotting analysis shows the protein expression of NR1D1, BMAL1, TOMM20, and LC3 II/I after cells were treated with mdivi-1 (12.5 μM, 25 μM, 50 μM, or a vehicle) for 48 h (n = 3). (**J**) Quantification of NR1D1, BMAL1, TOMM20, and LC3 II/I protein expression. * *p* < 0.05; ** *p* < 0.01; *** *p* < 0.001; **** *p* < 0.0001. Data are means ± S.E.M. Two-sided Student’s *t* tests were used to measure the significance.

**Figure 5 ijms-25-04853-f005:**
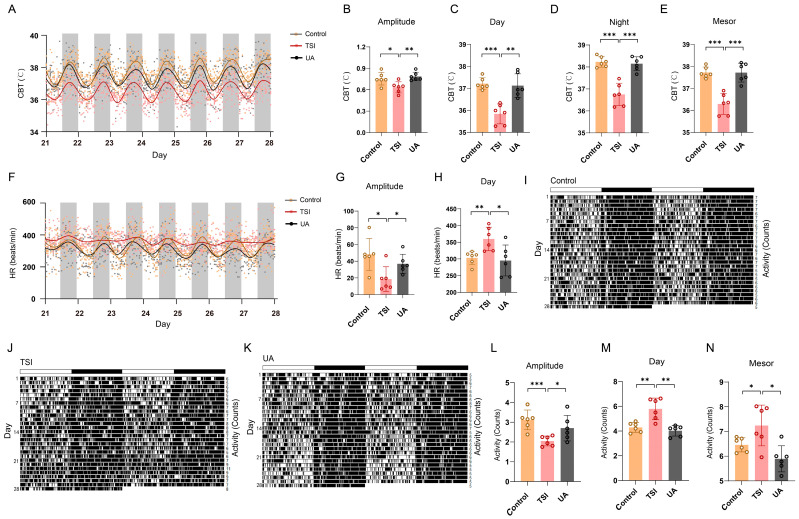
UA ameliorates the disturbance of SCN rhythms. (**A**) The curve fitting of the CBT data of rats during the final 7-day period from control, TSI, and UA-treated groups (each point represents the average CBT values corresponding to each hourly time point from control, TSI, and UA-treated rats). (**B**–**E**) Analysis of amplitude (**B**), daytime (**C**), night-time, (**D**) and mesor (**E**) of the CBT data during the final 7-day period. (**F**) The curve fitting of HR data of rats during the final 7-day period from control, TSI, and UA-treated groups (each point represents the average HR values corresponding to each hourly time point from control, TSI, and UA-treated rats). (**G**,**H**) Analysis of amplitude (**G**) and daytime (**H**) of the HR data during the final 7-day period. (**I**–**K**). Representative double-plotted actograms of locomotor activity of rats during the 28-day period from control, TSI, and UA-treated groups. (**L**–**N**) Analysis of amplitude (**L**), daytime (**M**), and mesor (**N**) of the locomotor-activity data during the final 7-day period. * *p* < 0.05; ** *p* < 0.01; *** *p* < 0.001. Data are means ± S.E.M., n = 6. Two-sided Student’s *t* tests were used to measure the significance.

**Figure 6 ijms-25-04853-f006:**
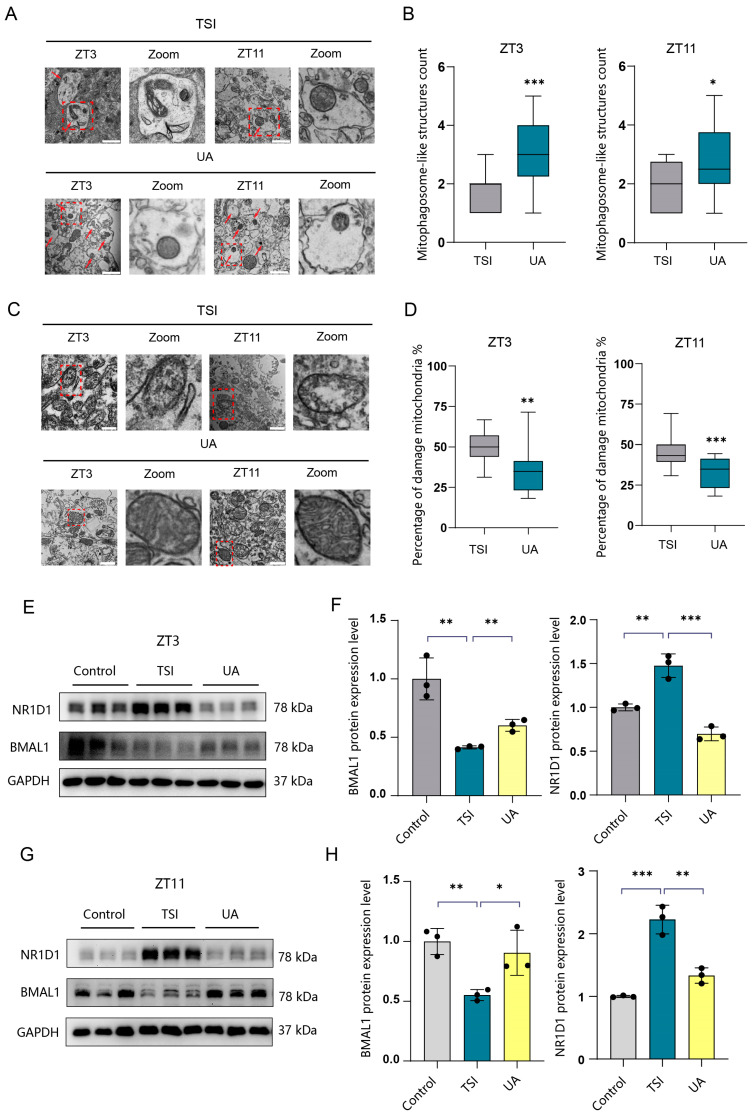
UA ameliorates the disturbance of circadian rhythms through promoting the degradation of NR1D1 in rats. (**A**) TEM images showing mitophagy in SCN neurons from TSI and UA-treated groups at ZT3 and ZT11 (n = 4). Mitophagosome-like structures are indicated by red arrowheads. The red dashed square indicates the zoom region. Scale bar, 1 μm. (**B**) Quantification of mitophagosome-like structures are indicated by red arrowheads. (**C**) TEM images displaying mitochondria in SCN neurons from TSI and UA-treated groups at ZT3 and ZT11 (n = 4). The red dashed square indicates the zoom region. Scale bars, 1 μm. (**D**) Quantification of the ratio of damaged mitochondria to total mitochondria. (**E**–**H**) Western blot analysis of NR1D1 and BMAL protein levels in SCN tissues from TSI and UA-treated groups at ZT3 and ZT11, followed by quantification of NR1D1 and BMAL protein expression (n = 3). * *p* < 0.05; ** *p* < 0.01; *** *p* < 0.001. Data are means ± S.E.M. Two-sided Student’s *t* tests were used to measure the significance.

**Table 1 ijms-25-04853-t001:** Sequences of primers used for RT-qPCR.

Gene Name	Forward Primer (5′-3′)	Reverse Primer (3′-5′)
*Nr1d1*	AGGTGACCCTGCTTAAGGCTG	ACTGTCTGGTCCTTCACGTTGA
*β-actin*	CCCTGGCTCCTAGCACCAT	GAGCCACCAATCCACACAGA
*COX II*	GATGACGAGCGACTGTTCCA	TGGTAACCGCTCAGGTGTTG
*Rpl13a*	GGTGGTGGTTGTACGCTGTGAG	CGAGACGGGTTGGTGTTCATCC

## Data Availability

The data that support the findings of this study are available from the corresponding author upon reasonable request.

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
