# Peer review of "Mitophagy Regulates the Circadian Rhythms by Degrading NR1D1 in Simulated Microgravity and Isolation Environments"

_ijms, 2024, doi:10.3390/ijms25094853_

Round 1
Reviewer 1 Report
Comments and Suggestions for Authors
The authors study the effects of tail suspension and isolation (TSI) on core body temperature (CBT), heart rate (HR) and locomotor activity rhythms. They find that TSI attenuates the rhythms of all three and then suggest a reduction in mitophagy in SCN neurons as the underlying cause of the attenuation. Although the rescue of CBT, HR and activity rhythms by administering mice with the benzo-coumarin urolithin a (UA) is a promising finding, I have several concerns regarding the study.
1. The authors use TSI as a model of space environment. TSI is a stress model that highly stresses the animals. What justifies calling the effects of TSI as simulating space environment? Unless the authors perform experiments that control for the stress the TSI imposes on the mice they cannot claim that TSI is a model of space environment. How do the authors know that the effects they observed in the TSI group are not driven by the stress the animals experience? I suggest restructuring the paper to discuss the study as a study of the effects of stress on rhythms and SCN neuron mitophagy.
2. The authors should provide the full set of data for the full 28 days for the experiment described in Figure 1. The authors show CBT data from days 21-28 to report that TSI reduces the amplitude of CBT rhythms. Depending on the data these might be supplemental figures but given the effect of the early period of the experiment on the final week for which we see data in figure 1, it is critical to see the full 28 day data. I appreciate that the authors show the representative activity plots but it is critical to show the data on activity for all mice (as with the CBT and HR (figures 1B, 1H respectively).
3. The manuscript gives the impression that NR1D1, BMAL1 are only degraded through autophagy, while existing literature shows that this is not true. The authors use MG132 to exclude proteasomal degradation of NR1D1, BMAL1. Are they suggesting that NR1D1, BMAL1 degradation happens only through autophagy?
4. How the 20mg/kg/d administration dose of UA compares to what is physiologically relevant? In other words, how do we know that this dose doesn’t have a supra-physiological effect? The authors should measure UA in tissues to see what concentrations were achieved by the administration. The authors should also make clear in Figure 5 the UA treatment period. Is the UA treatment 4 weeks long? Why the authors decide to show us only the last week of what seems like a 4-week experiment? The authors should provide the full set of data.
Minor comments
1. Page 2, line 90. Results 2.1 title: Please replace : “TSI environment disrupts the amplitude of circadian rhythms” with ”TSI environment reduces the amplitude of circadian rhythms”
2. The data on CBT rhythms (Figure 1B) are too variable to allow reliable calculation of Phase in CBT rhythms. The difference in phase reported is questionable. What is the statistic that was used and is it powered to detect a statistically significant difference?
Comments on the Quality of English LanguageThere are grammar mistakes that need to be corrected.
Reviewer 2 Report
Comments and Suggestions for Authors
The paper of Dr. Zhou, Dr. Li, Dr. Song, Dr. Qu, and coauthors, is using a rat model of simulated microgravity to explore the regulation of the circadian rhythms by mitophagy. The rat model is an asset for determining the molecular effects of microgravity on the health of mammals, before any long term space mission. The hypothesis derives from seminal observations reported by da Silveira et al (2020) on 59 astronauts and 2 mouse strains. The Nuclear Receptor Subfamily 1 Group D Member 1 (NR1D1) is playing a central role in the demonstration connecting mitophagy with the circadian clock in the neurons of suprachiasmatic nucleus. As such, the paper is adding a nice piece of information in this new field of research. The number of rats used in molecular demonstration is low (between 3 and 4 per group) but the experimental setting in chronobiology is always heavy to implement. The minimal number of rats has been used for meaningful statistical analyses.
I recommend the paper for publication, but I suggest adding in the discussion section, an information on the relation between circadian clock and mitophagy.
Major modification
The authors have performed a nice study on bmal1 / nr1d1, but the relation between the circadian clock and mitophagy has been already explored (Rabinovich-Nikitin I, Rasouli M, Reitz CJ, Posen I, Margulets V, Dhingra R, Khatua TN, Thliveris JA, Martino TA, Kirshenbaum LA. Mitochondrial autophagy and cell survival is regulated by the circadian Clock gene in cardiac myocytes during ischemic stress. Autophagy. 2021 Nov;17(11):3794-3812. doi: 10.1080/15548627.2021.1938913. Epub 2021 Aug 7. PMID: 34085589; PMCID: PMC8632283).
Did you explore the clock gene expression in parallel with bmal1 ? If not, the possibility of a more complex effect of microgravity on the circadian clock has to be underlined in the discussion section.
Reviewer 3 Report
Comments and Suggestions for Authors
The primary issue with this paper lies in the model utilized. Rather than accurately simulating microgravity as outlined, the methodology induces chronic stress and social isolation in rats. This is achieved by restraining the rats via tail suspension on a 30-degree plane for a duration of 28 days. Notably absent from the study is a description of the availability of food and water for the rats under these conditions. Additionally, it is concerning that several experiments were conducted with a sample size (n) of only 3, raising questions about the statistical robustness of the findings. Moreover, there is a lack of detail provided regarding the concentration and doses of agents administered both in vitro and in vivo, and positive and negative controls. These deficiencies in methodology undermine the validity and reliability of the study's conclusions.
Round 2
Reviewer 1 Report
Comments and Suggestions for Authors
Reply to response to comment 1: I appreciate that the authors measured ACTH and cortisol in blood to show that the animals are probably not stressed at day 28 of the experiment. This together with the change in the discussion of their data improves the paper.
Reply to response to comment 2: I appreciate that the authors show us the full 28 days of data for CBT and HR. However, I’m concerned by their response regarding the data (highlighted in yellow). Do they mean that the data in figures 1B, 1H are from 1 control rat and 1 TSI rat? Is that why they call the curve fittings “representative”? If that’s the case, there is a major issue with how the data are presented. The comparison of the averages in the rest of the figures simply is not enough. The authors should first generate the averages of each rat for a given time bracket (1 hour bins for example) and then make averages of those for the whole group (6 control animals, 6 TSI animals). Only this figure would be a faithful visual comparison between the rhythms in the two groups. As the data suggest that there is no major swift in the phase of the rhythms the figures are possible.
Comments on the Quality of English Language
No major issues found
Reviewer 3 Report
Comments and Suggestions for Authors
The author need to clarify the availability of food and water, ad libitum. Also concentration and dose need to be added in the Material and Methods
